# Dysregulated Metabolic Pathways in Subjects with Obesity and Metabolic Syndrome

**DOI:** 10.3390/ijms23179821

**Published:** 2022-08-29

**Authors:** Fayaz Ahmad Mir, Ehsan Ullah, Raghvendra Mall, Ahmad Iskandarani, Tareq A. Samra, Farhan Cyprian, Aijaz Parray, Meis Alkasem, Ibrahem Abdalhakam, Faisal Farooq, Abdul-Badi Abou-Samra

**Affiliations:** 1Qatar Metabolic Institute, Academic Health System, Hamad Medical Corporation, Doha, Qatar; 2Qatar Computing Research Institute (QCRI), Hamad Bin Khalifa University, Doha, Qatar; 3Department of Immunology, St. Jude Children’s Research Hospital, Memphis, TN 38104, USA; 4College of Medicine, QU Health, Qatar University, Doha, Qatar; 5Qatar Neuroscience Institute, Academic Health System, Hamad Medical Corporation, Doha, Qatar

**Keywords:** metabolomics, obesity, metabolic syndrome, inflammation, sphingomyelins

## Abstract

Background: Obesity coexists with variable features of metabolic syndrome, which is associated with dysregulated metabolic pathways. We assessed potential associations between serum metabolites and features of metabolic syndrome in Arabic subjects with obesity. Methods: We analyzed a dataset of 39 subjects with obesity only (OBO, *n* = 18) age-matched to subjects with obesity and metabolic syndrome (OBM, *n* = 21). We measured 1069 serum metabolites and correlated them to clinical features. Results: A total of 83 metabolites, mostly lipids, were significantly different (*p* < 0.05) between the two groups. Among lipids, 22 sphingomyelins were decreased in OBM compared to OBO. Among non-lipids, quinolinate, kynurenine, and tryptophan were also decreased in OBM compared to OBO. Sphingomyelin is negatively correlated with glucose, HbA1C, insulin, and triglycerides but positively correlated with HDL, LDL, and cholesterol. Differentially enriched pathways include lysine degradation, amino sugar and nucleotide sugar metabolism, arginine and proline metabolism, fructose and mannose metabolism, and galactose metabolism. Conclusions: Metabolites and pathways associated with chronic inflammation are differentially expressed in subjects with obesity and metabolic syndrome compared to subjects with obesity but without the clinical features of metabolic syndrome.

## 1. Introduction

Obesity, a condition in which the body accumulates excessive fat, has been strongly associated with various diseases, such as heart disease, diabetes, high blood pressure, joint disorders, and certain cancers. The World Health Organization (WHO) recently reported that the incidence of obesity has tripled since 1975 [1]. In 2016, the WHO reported more than 1.9 billion adults (18 years and above) were overweight, and over 650 million were obese [2]. Recent estimates show that every five units of higher body mass index (BMI) above 25 kg/m^2^ is associated with about 31% higher risk of premature death [3]. Obesity together with dysglycemia, dyslipidemia, and hypertension is called metabolic syndrome. Obesity is the main driver of metabolic syndrome, where its exponential rise has concomitantly resulted in an alarming incidence of metabolic disorder. Apart from metabolic syndrome, being overweight or obese predisposes people to a range of adverse health consequences, including endocrine disorders (e.g., advanced pubertal development and polycystic ovarian disease), cardiovascular disease (e.g., hypertension), respiratory symptoms including breathlessness and obstructive sleep apnea, and some malignancies [4]. Many recent studies [5,6,7] have indicated that metabolic syndrome is associated with increased risk of both atherosclerotic cardiovascular disease (ASCVD) and type 2 diabetes. Compared to normal persons, people with metabolic syndrome have at least a twofold increase in risk for ASCVD and about a fivefold risk for type 2 diabetes in both men and women [8]. In addition, diabetes is accompanied by microvascular disease, which is a common cause of chronic renal failure. The relationship between metabolic risk factors and development of ASCVD is complex and certainly not well understood.

Prevention and treatment of obesity and obesity-related diseases are therefore major public health challenges that need to be addressed. Obesity is a heterogeneous and complex condition, and the subgroup of individuals with obesity but no metabolic disorders has been described to have “only obesity” (OBO) [9]. In contrast to obesity with metabolic diseases (OBM), the OBO phenotype has a favorable lipid profile and normal or only slightly affected insulin sensitivity, despite the similar amount of body fat [9]. Better understanding of the mechanisms underlying obesity-related metabolic diseases and their resulting early complications, preferably before symptoms are evident, is crucial for developing new therapies.

Metabolomics is a technology for profiling and measuring the levels of low-molecular-weight metabolites (<1500 Da) in various systems, from cells to whole organisms [10]. Over the last decade, metabolomics studies have identified several relevant biomarkers involved in complex clinical phenotypes in diverse biological systems. Most diseases result in signature metabolic profiles that reflect the sum of external and internal cellular activity [11]. Untargeted metabolic characterization profiles all metabolites within a sample with the aim of identifying diverse metabolites to generate hypotheses. It can be used to identify biomarkers to unveil the molecular mechanisms of complex diseases, for monitoring diseases, and for risk evaluation.

Metabolic derangements between OBO and OBM have been explored, with initial findings suggesting that the OBO cohort is not at risk of metabolic morbidities such as cardiovascular disease (CVD) [9,12]. Other investigations found that risk of metabolic morbidities increases with obesity [13]. A recent systematic review performed a meta-analysis comparing OBO versus OBM and included 12 high-quality studies. The findings of the meta-analysis included shifts in the levels of branched-chain amino acids, aromatic amino acids, and acylcarnitine. These biomarkers are very similar to those identified in patients diagnosed with coronary heart disease, suggesting a high incidence of cardiovascular disease in the OBM group [14]. Moreover, the risk of cardiovascular diseases increases with age; therefore, age can influence the OBM group. There is need to investigate the metabolic differences between OBO and OBM groups while controlling for factors such as comorbidities and age.

The rationale of this study was to identify the underlying metabolic pathways and metabolic mechanisms that are differentially regulated in OBO vs. OBM subjects. Even though the concept of metabolically healthy obesity remains controversial, a profound understanding of the underlying metabolic regulation between the OBO and OBM is necessary to enhance the current knowledge of development and regulation pathways and to optimize prevention and treatment strategies. In the current study, we aimed to discover metabolic variations in obese individuals with and without metabolic disorders. The criteria for group assignment were based on metabolic syndrome, obesity, and any two of the following conditions: triglycerides ≥ 150 mg/dL (1.7 mmol/L), HDL < 40 mg/dL (1.03 mmol/L) in men or <50 mg/dL (1.29 mmol/L) in women, blood pressure ≥ 130/85 mm Hg, and fasting blood glucose ≥ 110 mg/dL (5.6 mmol/L) [15]. The individuals in the OBO and OBM groups were age-matched. We used pathway enrichment analysis and statistical analysis to identify individual metabolites that have different levels in OBO compared to OBM. Lysine degradation, amino sugar and nucleotide sugar metabolism, arginine and proline metabolism, fructose and mannose metabolism, and galactose metabolism were the significantly enriched pathways (*p* < 0.05). We identified 83 metabolites that were significantly different between OBO and OBM (*p* < 0.05). The identified biomarkers were mostly lipids and amino acids. Most of the lipids that had low concentrations in OBM were associated with inflammation, indicating a key role of inflammation in differentiating OBO and OBM groups.

## 2. Results

### 2.1. Baseline Characteristics of the Study Population

The participants were divided into OBO and OBM groups. The participants in the two groups were age-matched, resulting in 39 subjects (18 in OBO and 21 in OBM) included in the study.

The baseline characteristics of the participants are summarized in Table 1. The mean age of OBO and OBM groups is 38.06 ± 4.21 and 40.52 ± 7.26 years, respectively, and the BMI is 40.95 ± 4.48 and 39.64 ± 2.90 kg/m^2^, respectively. The number of females in the OBO group is more than those in the OBM group, but it is not statistically significant (*p* = 0.415).

Mean values of ALT, AST, and albumin are significantly different between the genders within the groups and are higher in males compared to females, reflecting the normal trend. Mean values of cholesterol and creatinine are significantly higher in males compared to females in OBO, whereas the mean value of CRP is significantly lower in males compared to females in OBO.

Mean values for glucose, HDL, glycosylated hemoglobin HbA1C, and triglycerides are baseline clinical parameters that exhibit statistically significant differences between the OBO and OBM groups. Similar trends in the mean values for these clinical parameters are found in the same-gender participants across the groups, although they are not statistically significant for one of the genders.

### 2.2. Univariate Analysis

Univariate analysis of the metabolomics profile was performed using logistic regression. Out of the 696 metabolites analyzed, 83 metabolites were found in significantly different levels (*p* < 0.05) in the OBM group relative to the OBO group. The fold-change in the concentration of metabolites is plotted against the *p*-value in the volcano plot in Figure 1A. The number of metabolites significantly changed in super classes is summarized in Figure 1B. Among the significant metabolites, 66 (79%) metabolites were decreased in OBM, including lipids (*n* = 35), amino acids (*n* = 15), peptides (*n* = 6), nucleotides (*n* = 4), partially characterized molecules (*n* = 3), carbohydrates (*n* = 2), and cofactors/vitamins (*n* = 1). Of the remaining, 17 metabolites were elevated in OBM, including lipids (*n* = 8), carbohydrates (*n* = 3), amino acids (*n* = 3), peptides (*n* = 1), xenobiotics (*n* = 1), and energy-related metabolites (*n* = 1).

In the following subsections, we discuss some metabolites of interest in detail. The metabolites selected are highly significant and are directly or indirectly related to inflammation.

#### 2.2.1. Sphingomyelins Are Significantly Decreased in OBM

The most significant differences among metabolites (*p* < 0.05) in OBM compared to OBO were lipids. Interestingly, several sphingomyelins were significantly decreased in OBM compared to OBO (Figure 2). The sphingolipids include hydroxypalmitoyl sphingomyelin (d18:1/16:0(OH))** (*p* = 0.007), palmitoyl sphingomyelin (d18:1/16:0) (*p* = 0.008), sphingomyelin (d17:1/14:0, d16:1/15:0)* (*p* = 0.041), sphingomyelin (d17:1/16:0, d18:1/15:0, d16:1/17:0)* (*p* = 0.01), sphingomyelin (d17:2/16:0, d18:2/15:0)* (*p* = 0.004), sphingomyelin (d18:1/17:0, d17:1/18:0, d19:1/16:0) (*p* = 0.013), sphingomyelin (d18:1/18:1, d18:2/18:0) (*p* = 0.004), sphingomyelin (d18:1/19:0, d19:1/18:0)* (*p* = 0.017), sphingomyelin (d18:1/20:1, d18:2/20:0)* (*p* = 0.002), sphingomyelin (d18:1/20:2, d18:2/20:1, d16:1/22:2)* (*p* = 0.014), sphingomyelin (d18:1/21:0, d17:1/22:0, d16:1/23:0)* (*p* = 0.046), sphingomyelin (d18:1/22:1, d18:2/22:0, d16:1/24:1)* (*p* = 0.006), sphingomyelin (d18:1/22:2, d18:2/22:1, d16:1/24:2)* (*p* = 0.003), sphingomyelin (d18:1/24:1, d18:2/24:0)* (*p* = 0.006), sphingomyelin (d18:2/14:0, d18:1/14:1)* (*p* = 0.04), sphingomyelin (d18:2/16:0, d18:1/16:1)* (*p* = 0.007), sphingomyelin (d18:2/18:1)* (*p* = 0.006), sphingomyelin (d18:2/21:0, d16:2/23:0)* (*p* = 0.029), sphingomyelin (d18:2/23:0, d18:1/23:1, d17:1/24:1)* (*p* = 0.003), sphingomyelin (d18:2/24:1, d18:1/24:2)* (*p* = 0.003), sphingomyelin (d18:2/24:2)* (*p* = 0.005), and stearoyl sphingomyelin (d18:1/18:0) (*p* = 0.025).

In the box plots, the upper and lower ends of the box represent the 25th and 75th percentiles of the data, respectively. The whiskers on the top and bottom of the box represent data above the 75th percentile and below 25th percentile, respectively. The line in the middle of the box represents the median of the data. The data shown in the box plots are normalized with zero mean and unit variance. The data were normalized to make comparison between different compounds easier when the plots are shown side-by-side. The same convention is used for the rest of the box plots.

#### 2.2.2. Quinolinate Is Significantly Decreased in OBM

Quinolinate was the only significant (*p* = 0.006) cofactor found in the panel of metabolites differentially expressed in OBO versus OBM. The concentration of quinolinate, along with kynurenine (*p* = 0.097) and tryptophan (*p* = 0.308), was found to be low in OBM compared to OBO subjects (Figure 3).

### 2.3. Pathway Enrichment Analysis

The results of metabolic pathway enrichment analysis are shown in Table 2. The pathways shown have *p* < 0.05. False discovery rate (FDR) correction was not used due to the small sample size. The enriched pathways include lysine degradation (enrichment ratio (ER) = 4.18), amino sugar and nucleotide sugar metabolism (ER = 5.09), arginine and proline metabolism (ER = 2.21), fructose and mannose metabolism (ER = 3.91), and galactose metabolism (ER = 3.07).

The KEGG pathway map for arginine and proline metabolism is shown in Figure 4. The colored nodes represent metabolites found in our metabolomics panel. The metabolites are highlighted based on the mean concentration in OBM and OBO groups, where blue, yellow, and red represent low, same, and higher mean concentrations of metabolites, respectively, in OBM compared to OBO. A high-resolution map is available in Appendix A. Among the 12 compounds in arginine and proline metabolism that were present in our metabolomic profile, guanidinoacetate and pyruvate had higher concentrations in OBM compared to OBM, whereas *n*-acetylputrescine and N4-acetamidobutanoate had lower concentrations in OBM compared to OBO. The higher concentration of pyruvate cannot be completely attributed to arginine and proline metabolism, as it is also produced by metabolism of sugars. The changes in the ratio of ornithine and L-arginine in OBM and OBO indicates a shift in the metabolism of L-arginine to produce nitric oxide (NO). The ratio of mean concentration of ornithine and L-arginine in OBO is 0.88, whereas the ratio is 1.01 in OBM. The increased ratio indicates less production of nitric oxide.

### 2.4. Association of Metabolite Concentration with Clinical Parameters

Association of metabolites differentially expressed in OBO and OBM with clinical features of metabolic syndrome was also investigated using Pearson correlation. The clinical parameters included glucose, HbA1C, insulin, triglycerides, albumin, ALT, AST, HDL, LDL, cholesterol, and CRP. The heatmap of calculated correlations is shown in Figure 5. The *p*-values for the correlations were also computed using the ‘cor.test’ function in the stats package of R. A cross placed on the correlation cell in the heatmap indicates no significance (*p* < 0.05).

Glucose, lactate, fructose, mannitol/sorbitol, mannose, proline, and *n*-acetyleneuraminate were found positively correlated with the clinical laboratory values of glucose and HbA1C. Arginine and N6-acetyllysine were negatively correlated with glucose and HbA1C. Ornithine was positively correlated with HbA1C, whereas quinolinate, sphingomyelin (d18:2/18:1)*, sphingomyelin (d18:1/24:1, d18:2/24:0)*, and sphingomyelin (d18:2/24:1, d18/24:2)* were negatively correlated with HbA1C.

Sphingomyelin (d18:2/21:0, d16:2/23:0)*, hydroxypalmitoyl sphingomyelin (d18:1/16:0(OH))**, spermidine, sphingomyelin (d18:2/23:0, d18:1/23:1, d17:1/24:1)*, sphingomyelin (d18:1/17:0, d17:1/18:0, d19:1/16:0), sphingomyelin (d18:1/22:1, d18:2/22:0, d16:1/24:1)*, sphingomyelin (d18:1/21:0, d17:1/22:0, d16:1/23:0)*, sphingomyelin (d18:1/24:1, d18:2/24:0)*, and sphingomyelin (d18:2/24:1, d18:1/24:2)* were negatively correlated with insulin, whereas pyruvate was positively correlated.

Several sphingomyelins were positively correlated with HDL, LDL, and cholesterol. Positive correlation with HDL was also found with deoxycarnitine, quinolinate, 4-guanidinobutanoate, and N6,N6,N6-trimethyllysine. Moreover, glycerol was negatively correlated with LDL.

## 3. Discussion

We used untargeted metabolomics to profile metabolites in individuals who are only obese as opposed to obese with metabolic diseases, and we identified enriched metabolic pathways and metabolites with significantly different concentrations in the two groups. The enriched pathways include lysine degradation, amino sugar and nucleotide sugar metabolism, arginine and proline metabolism, fructose and mannose metabolism, and galactose metabolism. Lysine is degraded through two pathways: via formation of saccharopine and via the pipecolic acid pathway. Lysine degradation is mainly done in the mitochondria, and tissue-specific roles of lysine degradation pathways are still not well-studied [16]. Amino sugar and nucleotide sugar metabolism pathways are enriched as seen by higher concentrations of glucose, mannose, and fructose in OBM vs. OBO groups. Although changes in arginine and ornithine are not statistically significant, the ornithine to arginine ratio in OBM (1.01) is higher than in OBO (0.88), indicating a shift in nitric oxide production for cardiovascular functions [14]. Enrichment of fructose and mannose metabolism and galactose metabolism pathways is an indicator of diabetes [17,18,19] in the OBM group, which corroborates the clinical data.

Pathway enrichment analysis is dependent on the presence of metabolites in a specific pathway in the metabolomic panel. Therefore, identification of the importance of a pathway cannot solely rely on pathway enrichment analysis. Changes in individual metabolites can also play a significant role as potential biomarkers. In our results, we identified 22 sphingomyelins decreased in OBM compared to OBO (*p* < 0.046). Sphingomyelin is a type of sphingolipid that is found in cell membranes of animals and constitutes around 85% of all sphingolipids [20]. Sphingomyelin breakdown is known to trigger multiple signaling pathways with outcomes as diverse as cell proliferation, differentiation, growth-arrest, and apoptosis [21]. A higher concentration of sphingomyelins in plasma is considered a risk factor for coronary heart disease [22]. Moreover, sphingomyelins play a major role in inflammatory diseases [23]. In a previous study, sphingomyelins were found to be related to obesity measures [24]. It is suspected that in sphingolipids, the saturation level of fatty acids and the length of the carbon chain may cause insulin resistance [25,26]. Higher plasma sphingomyelin levels have also been found in obese people [24,27]. Moreover, reduced metabolism of sphingolipids plays a role in the prognosis of type 2 diabetes and is linked to pancreatic β cell dysfunction [28].

Quinolinate is the only cofactor found significantly different in OBM compared to OBO. Quinolinate is an essential metabolite that plays an important role in the de novo synthesis of nicotinamide adenine dinucleotide (NAD+) from tryptophan in the kynurenine pathway [29]. In response to inflammation, quinolinate can increase kynurenine and cellular NAD+ levels [29]. Based on our results, we suspect the OBO group has a higher level of inflammation (higher CRP value in Table 1).

Association analysis of metabolites with clinical parameters revealed negative correlation between sphingomyelin and glucose, HbA1C, insulin, and triglycerides. Positive correlation between sphingomyelin and HDL, LDL, and cholesterol was found, which verifies earlier research [30].

In this study, obese subjects without clinical features of metabolic syndrome were compared with obese subjects with metabolic conditions. The results identified metabolic derangements in the subjects with obesity and metabolic syndrome. The present study found biomarkers and pathways that have previously been associated with obesity and chronic inflammation but not investigated from the perspective of obesity with metabolic disorders. In contrast to a recent study [31], we have a slightly larger sample size and more stringent participant-inclusion criteria.

There are a few limitations of the current study. Due to age- and BMI-matching, the number of participants in our study is small. The small number of participants reduces the statistical power of the analysis. Moreover, causal relationships between different factors cannot be determined using a cross-sectional study.

In conclusion, metabolites and pathways associated with chronic inflammation are differentially expressed in subjects with obesity and metabolic syndrome compared to subjects with obesity but without the clinical features of metabolic syndrome. We believe that further investigations can explore the mechanistic changes in the metabolites and pathways over time to identify potential therapeutic interventions to prevent the onset of metabolic disorders in these populations.

## 4. Materials and Methods

### 4.1. Study Population

The institutional review board (IRB) of Hamad Medical Corporation approved the study protocol (IRB protocol #16245/16). All participants were recruited at Qatar Metabolic Institute. An informed consent form was available in both English and Arabic. Study coordinators discussed verbally the details of the study with the participants when they were given the written consent form. Participants who voluntarily decided to take part in the study signed the consent form for recruitment in the study. The participants comprised obese adults of both genders with BMI ≥ 35 kg/m^2^ who had no other chronic illnesses or terminal conditions. Fasting blood samples were taken from the participants. Venous blood samples were collected in serum separator tubes, centrifuged at 1200× *g* at 4 °C for 10 min, aliquoted to avoid any further freeze–thaw cycles, and then stored at −80 °C until the time of measurements.

### 4.2. Baseline Statistical Analysis

Baseline statistical analysis was performed to compute the means, standard deviations, and *p*-values of study participants for the OBO and OBM groups. A very small fraction of values in some clinical parameters were missing (the total number of missing values across all parameters was 1.88%). The missing values were replaced with the median of the corresponding variables. To compute *p*-values, different tests were used based on the type and distribution of the values. The chi-square test was used for categorical variables. The Wilcoxon test was performed for variables not following normal distribution for continuous variables. The normality of variables was tested using the Shapiro–Wilk test. The Student’s *t*-test was used for variables with normal distribution.

### 4.3. Metabolomics Profiling and Quality Control

Untargeted, ultrahigh-performance liquid chromatography–tandem mass spectroscopy (UPLC–MS/MS) was performed and curated by Metabolon on serum metabolites for OBO and OBM jointly. Batch-normalized data were generated by normalizing samples across batches to correct for minor instrument variations by scaling each batch’s medians to one and scaling the rest of the data points proportionally. A total of 1069 metabolites were profiled within ten super classes: lipids (*n* = 360, 33.68%), amino acids (*n* = 203, 18.99%), xenobiotics (*n* = 169, 15.81%), peptides (*n* = 47, 4.40%), nucleotides (*n* = 38, 3.55%), cofactors and vitamins (*n* = 31, 2.90%), carbohydrates (*n* = 20, 1.87%), partially characterized molecules (*n* = 17, 1.59%), energy (*n* = 10, 0.94%), and unknown (*n* = 174, 16.28%). Unknown metabolites were removed, leaving 895 metabolites for further analysis.

Data were preprocessed to ensure data quality. Metabolites and samples with more than 20% missing data were removed based on the criteria suggested by Wei et al. [32]. A total of 199 metabolites were removed, leaving 696 metabolites. Data imputation was performed by replacing missing values for each metabolite in the data with the minimum value detected for the metabolite. Principal component analysis (PCA) was performed to identify sample outliers. A sample was considered an outlier if its first five principal component values fell outside (µ ± 3SD). One OBO and one OBM sample were removed (Appendix A). Winsorization was performed to reduce the impact of outliers in the metabolites. Values below the 10th percentile were replaced with the 10th percentile, and values above the 90th percentile were set to the 90th percentile for each metabolite. After quality control, 696 metabolites and 37 samples (17 OBO and 20 OBM) remained for subsequent analysis (Appendix A).

### 4.4. Univariate Statistical Analysis

Differences between individual metabolites were determined using logistic regression. The analysis was adjusted for age, sex, and BMI to mitigate their impact on the respective metabolites. False discovery rate (FDR) controlling was not performed due to the small sample size. Logistic regression was performed using the ‘glm’ function in the stats package of R (R Core Team, version 4.1.1, 2021, Vienna, Austria) to compute the *p*-value and the effect size. The effect determines the direction of change in the metabolite concentration between OBO and OBM: a positive effect size means a higher concentration in OBM compared to OBO and vice versa.

### 4.5. Pathway Enrichment Analysis

Pathway enrichment analysis was performed using MetaboanalystR v3.0 [33] with HMDB ids provided by Metabolon to match the compounds in the metabolite sets. Data were normalized using the auto-normalization option, which transforms data to zero mean and unit variance. Quantitative enrichment analysis (QEA) was used, employing metabolite concentrations to develop a generalized linear model for estimation of the Q-stat (a statistic) of the metabolite set. Q-stat is a measure of the correlation between compound concentrations and the given phenotype [34]. Metabolite sets having a significant change in concentration in only a few compounds, or many compounds having small, correlated changes are identified by this approach. The enrichment ratio is the ratio of Q-stat for the given data to its expected value by chance. An ER greater than one for a metabolite set indicates different metabolite concentrations in the set.

KEGG pathway maps were annotated for visualization using the online tool Pathview (https://pathview.uncc.edu/) [35,36]. Pathview annotated the KEGG pathway maps by filling the metabolite nodes with colors representing the ratio of normalized metabolite concentrations of OBM to OBO. Red, yellow, and blue colors represent higher, same, and lower concentrations of metabolites in OBM compared to OBO, respectively.

## Figures and Tables

**Figure 1 ijms-23-09821-f001:**
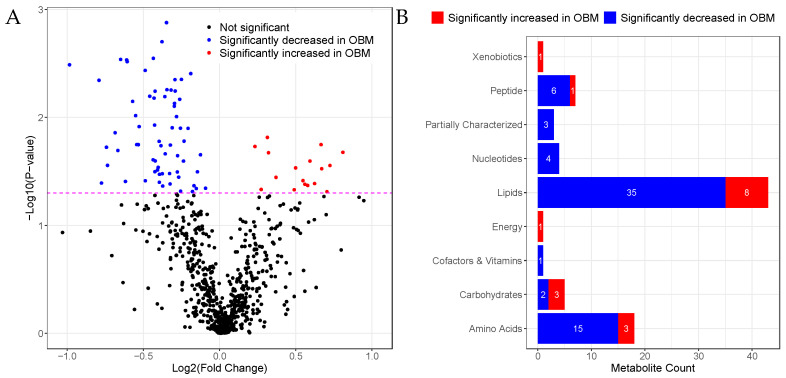
Changes in metabolite concentrations in OBM vs. OBO. (**A**) Volcano plot of log2 (fold-change) versus −log10(*p*-value) of metabolites. Metabolites above the dashed line have *p* < 0.05. (**B**) Bar plot showing the number of metabolites significantly increased or decreased in a super pathway.

**Figure 2 ijms-23-09821-f002:**
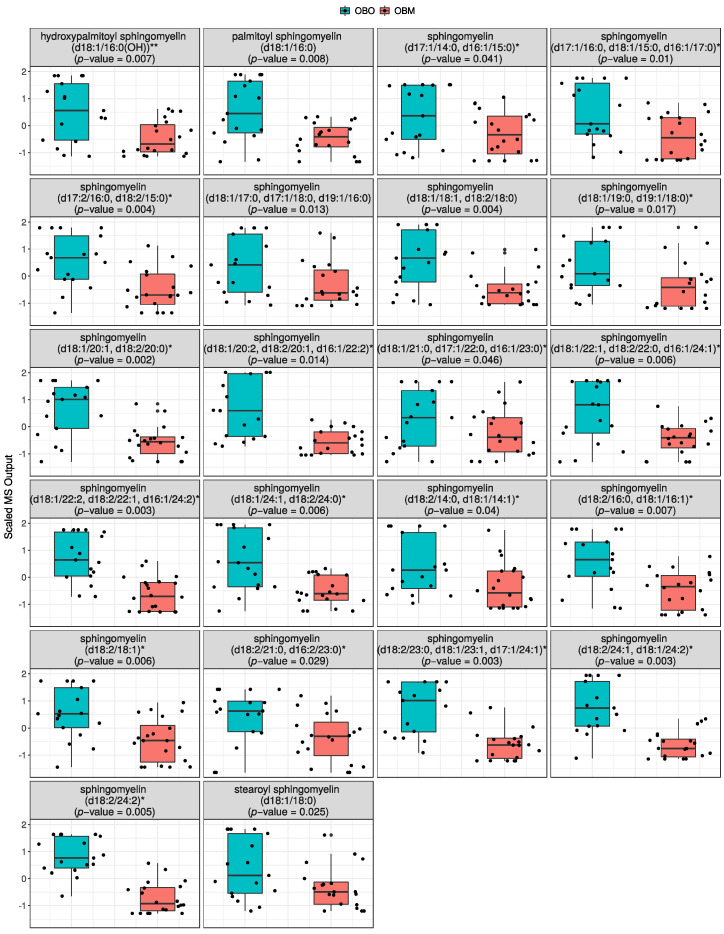
Distribution of statistically significant sphingomyelins in OBO (**left** groups) and OBM (**right** groups). The box plots are supplemented with violin plots to show distribution of samples across different MS values of the sphingomyelins.

**Figure 3 ijms-23-09821-f003:**
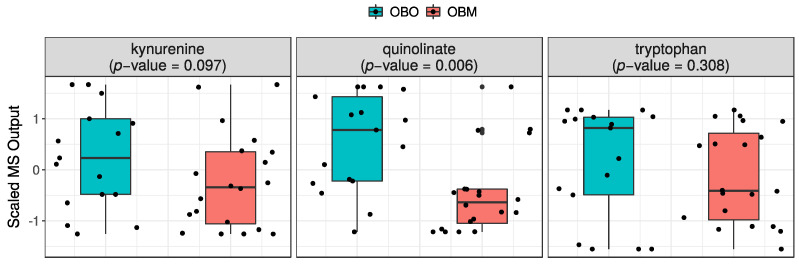
Distribution of kynurenine, quinolinate, and tryptophan in OBO (**left** groups) and OBM (**right** groups). The box plots are supplemented with violin plots to show distribution of samples across different MS values of the sphingomyelins.

**Figure 4 ijms-23-09821-f004:**
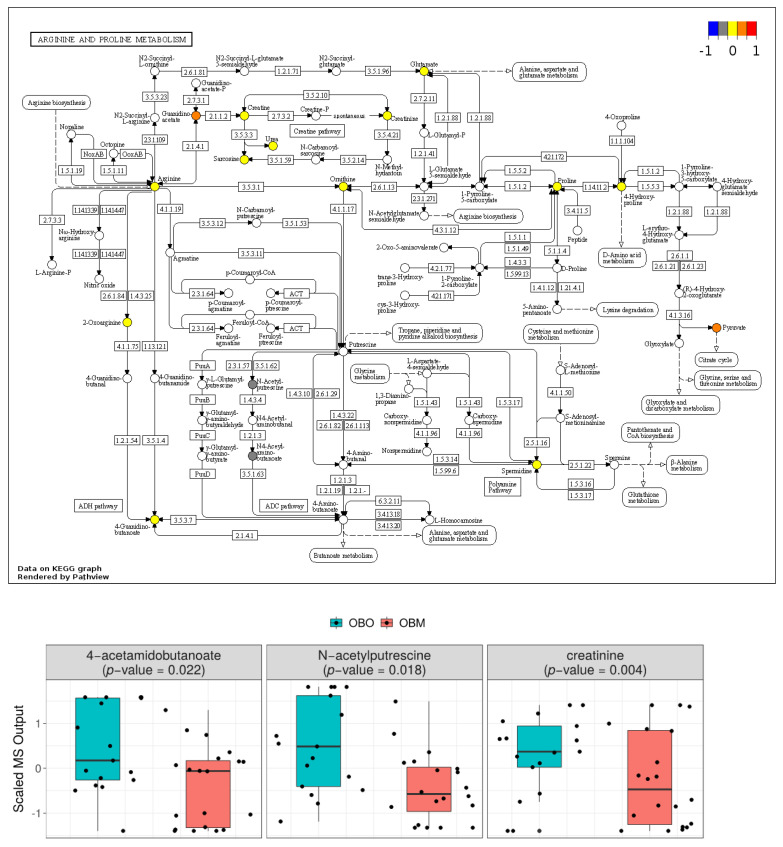
KEGG pathway map of arginine and proline metabolism. Metabolites in blue, yellow, and red have lower, the same, and higher concentrations, respectively, in OBM compared to OBO. Non-colored metabolites were not part of the metabolomics profile. The box plots show distribution of significant metabolites (*p* < 0.05) in the pathway. The box plots are supplemented with violin plots to show distribution of samples across different MS.

**Figure 5 ijms-23-09821-f005:**
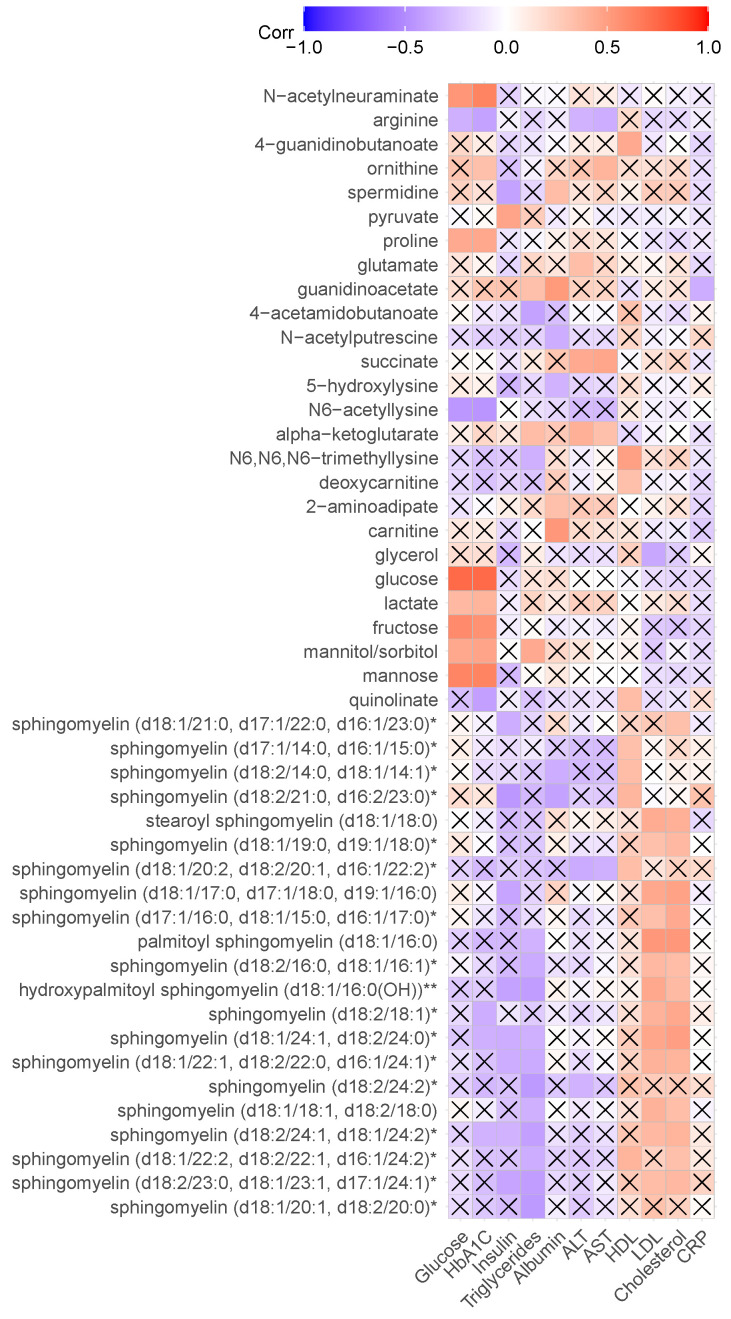
Heatmap of correlation of differentially expressed metabolites in OBO and OBM. Each row represents a metabolite, and a column represents a clinical parameter. A cross indicates that correlation is not significant (*p* < 0.05).

**Table 1 ijms-23-09821-t001:** Cohort characteristics with reported mean (standard deviation) for all variables: *p*-values correspond to differences between the means; underlined *p*-values are significantly below the threshold of 0.05.

	OBO				OBM				OBO vs. OBM		
	M *n* = 7	F *n* = 11	All *n* = 18	*p_g_*	M *n* = 12	F *n* = 9	All *n* = 21	*p_g_*	*p_m_*	*p_f_*	*p_a_*
ALT	28.1 (12.1)	15.9 (8.8)	20.7 (11.6)	0.026	48.8 (42.5)	20.1 (8.8)	36.5 (35.1)	0.042	0.137	0.310	0.063
AST	25.8 (12.6)	14.3 (2.5)	18.8 (9.6)	0.006	28.8 (18.0)	16.7 (4.8)	23.6 (15.0)	0.044	0.712	0.337	0.251
Age	39.9 (3.0)	36.9 (4.6)	38.1 (4.2)	0.153	41.3 (7.8)	39.6 (6.8)	40.5 (7.3)	0.497	0.588	0.402	0.283
Albumin	40.8 (3.7)	35.8 (3.9)	37.7 (4.5)	0.030	40.7 (2.8)	36.9 (2.3)	39.1 (3.2)	0.008	0.866	0.939	0.367
BMI	38.6 (4.0)	40.9 (4.7)	40.0 (4.5)	0.238	40.4 (2.4)	38.7 (3.3)	39.6 (2.9)	0.201	0.271	0.196	0.746
C-Peptide	4.9 (5.2)	3.2 (1.4)	3.9 (3.4)	0.429	4.5 (1.7)	3.5 (1.0)	4.1 (1.5)	0.111	0.847	0.605	0.814
CRP	9.0 (6.5)	14.0 (13.1)	12.0 (11.0)	0.364	5.3 (3.1)	10.6 (2.8)	7.6 (4.0)	0.003	0.218	0.428	0.121
Cholesterol	5.7 (1.2)	4.5 (0.8)	4.9 (1.1)	0.025	4.8 (1.3)	4.8 (0.9)	4.8 (1.1)	0.887	0.154	0.437	0.665
Creatinine	80.9 (7.7)	59.0 (9.8)	67.5 (14.1)	<0.001	70.8 (14.0)	58.1 (11.2)	65.3 (14.1)	0.059	0.057	0.820	0.563
Glucose	5.1 (0.7)	5.2 (0.5)	5.2 (0.6)	0.819	6.5 (1.4)	8.5 (4.9)	7.3 (3.4)	0.268	0.031	0.081	0.009
HDL	1.6 (1.0)	1.4 (0.4)	1.5 (0.7)	0.766	0.9 (0.4)	1.1 (0.1)	1.0 (0.3)	0.275	0.166	0.006	0.008
HbA1C	5.6 (0.3)	5.5 (0.3)	5.5 (0.3)	0.383	6.8 (1.2)	7.3 (2.6)	7.0 (1.9)	0.618	0.016	0.068	0.002
Insulin	16.2 (7.1)	21.0 (16.1)	19.1 (13.3)	0.479	30.4 (15.1)	22.9 (8.3)	27.2 (12.9)	0.277	0.056	0.743	0.062
LDL	3.4 (1.8)	2.4 (0.7)	2.8 (1.3)	0.179	2.6 (1.3)	2.6 (0.9)	2.6 (1.1)	0.915	0.397	0.492	0.728
Triglycerides	1.5 (0.4)	1.3 (0.5)	1.4 (0.5)	0.318	2.8 (1.8)	2.4 (1.1)	2.7 (1.5)	0.523	0.031	0.013	0.001

M: males; F: females; All: males and females in the group; *p_g_*: *p*-value corresponding to differences between the means for males and females within the group; *p_m_*: *p*-value corresponding to differences between the means for males across the groups; *p_f_*: *p*-value corresponding to differences between the means for females across the groups; *p_a_*: *p*-value corresponding to differences between the means for all, across the groups.

**Table 2 ijms-23-09821-t002:** Pathway enrichment analysis of metabolites. Only pathways with *p* < 0.05 are reported. Total represents the number of metabolites identified by KEGG in that pathway, and hits represents the number of metabolites associated in the present study.

Pathway	Total	Hits	Statistic Q	Expected Q	*p*
Lysine degradation	25	5	11.615	2.778	0.003
Amino sugar and nucleotide sugar metabolism	37	3	14.145	2.778	0.005
Arginine and proline metabolism	38	12	6.149	2.778	0.015
Fructose and mannose metabolism	20	3	10.852	2.778	0.017
Galactose metabolism	27	6	8.527	2.778	0.020

## Data Availability

Not applicable.

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
