# Peer review of "Dysregulated Metabolic Pathways in Subjects with Obesity and Metabolic Syndrome"

_ijms, 2022, doi:10.3390/ijms23179821_

Round 1
Reviewer 1 Report
The authors have studied the metabolic variations in obese individuals with and without metabolic disorders. Paper is fairly written and can be considered subject to certain revisions. As mentioned below:
1. The rationale is not clear about the study. Precise reason as to why the study was carried out needs to be mentioned in the introduction.
2. Authors mention “The criteria for group assignment were based on metabolic syndrome, obesity and 225 any two of the following conditions”. Which classification was used for this; is it as per IDF or other. Please clarify with appropriate referencing.
3. Details of informed consent process needs to be included.
4. The details of blood collection, whether it is fasting or non-fasting and other details are missing.
5. Authors should include a paragraph on limitations of current study in the discussion section.
6. At the end of discussion, there is a paragraph: “Authors should discuss the results and how they can be interpreted from the perspective of previous studies and of the working hypotheses. The findings and their implications should be discussed in the broadest context possible. Future research directions may also be highlighted.” What is this paragraph referring to?
7. Conclusions of the study and future perspective should be provided at the end of the discussion section.
8. Language needs revision.
Reviewer 2 Report
The idea of comparing obesity as a part of metabolic syndrome with plain obesity is fine and present in much current literature, although there are not many non-focused metabolomics studies to study differences as the one done here. The present study is very extensive concerning the metabolites studied, with state-of-the art metabolomics, using mass spectrometry as the detection/quantification technique. The sample size is still small, although larger than in the other quoted study. The results obtained are valuable although they fail to identify causal relations that might explain why in some persons obesity is linked to metabolic syndrome and in other cases it is not linked to metabolic syndrome. The inflammatory component of the metabolic syndrome has been recognized already for some years (and this is not adequately referenced; perhaps you could refer to some review study such as the one in DOI: 10.1007/978-3-319-48382-5_7), so the conclusions are not spectacular, but nevertheless are valuable and add up valuable and well-collected data.
I do not feel sufficiently qualified for judging in deep the numerous data manipulations and statistical analyses that may be involved in a metabolomics paper as this is. I would recommend a more user-friendly approach. In particular:
1. I do not like the violin-type plots superimposed on the boxplots. Since the number of data is not so high for each group, it would be better to represent the actual points of the data (box plot with data superimposed) instead of showing the abstract concept of a probability density (truly an inference rather than the actual data), which is what is represented by the violin plot. Furthermore, boxplots are not universal in the way that they are represented, so please define in the first appearance what is shown by the lower and upper whiskers, the upper and lower ends of the box (75 and 25 percentiles?) and the horizontal line (mean or median?). Similarly, what is the meaning of the ordinate title “Scaled MS output”? Why not give the absolute value in terms of mg/L or mM or something like that? If there is some reason for not giving an absolute value for each parameter, please explain the meaning of the ordinate values.
2. The KEGG part of figure 4 is low resolution and difficult to follow. Please do provide a better resolution figure.
3. Please begin the Results section by describing the groups and their baseline differences, indicating the number of subjects in each groups and the type of differences being compared.
4. I don’t agree with the formulation in lines 88 and 89. I would change it to “Out of the 696 metabolites analyzed, 83 metabolites were found in significantly different levels (p <0.05) in the OBM group relative to the OBO group.”
5. Differences between frequency of females in the two groups has not a p= 0.063 as indicated in line 264, but p=0.415 (Table 2)
6. Please indicate in Table 2 that the p values correspond to differences between the means for the “all” columns.
7. The statistical analysis in Table 2 is incomplete, since it should include the comparison of the means between males and females within each group, and between the effectives of the same sex in the two groups.
8. Furthermore, what means the sentence (lines 243-244) “The missing values (1.88%) in clinical parameters were replaced with the median of the variables”? Does it mean that n values differ for the different parameters of the table from the n values given in the first line of the table? If this is so, please give n values for each parameter for each column of the Table. In this way, each value would be defined by three values, mean, standard deviation, and n value.
9. The definition of the values in parentheses in Table 2 is confusing, as in the first line it represents percentage, while in the other lines it is the Standard deviation.
10. The sentence (lines 235-236) “Creatinine, glucose, HDL and triglycerides are among the clinical parameters that are statistically significant” is not well expressed and it is wrong, as it should read “Mean values for glucose, HDL, glycosylated hemoglobin HbA1C and triglycerides are baseline clinical parameters that exhibit statistically significant differences between the OBO and OBM groups”.
11. What means (lines 275-276) that “The analysis was adjusted for age, sex and BMI to mitigate their impact on the respective metabolites.”? The number of samples appears too little to do such adjustment.
12. Please remove lines 213-216. They are instructions from the journal
13. Please give the correct supplementary material information in lines 299-300
Reviewer 3 Report
This paper entitled ‘Dysregulated metabolic pathways in subjects with obesity and metabolic syndrome’ demonstrates that metabolites and pathways associated with chronic inflammation are differentially expressed in obese subjects with metabolic syndromes compared to obese subjects without the clinical features of the metabolic syndrome. I have the following questions:
1. Does gender matter the differential pattern in obese subjects with/without metabolic syndromes?
2. Please define the OBM specifically in Introduction section.
3. The data interpretation for Fig.2 is basically missing. Please describe each panel statistically and specifically.
4. Data interpretation for KEGG analysis is not equal to its fig legend. Please refine the respective writing for KEGG analysis.
5. Once a phrase is defined, the following use of this concept should be unified as the abbreviation.
6. What does the final paragraph mean in Discussion section? I’m confused.
7. The sub headings for “Materials and Methods” should be 4.1, 4.2, 4.3…
Round 2
Reviewer 1 Report
All the suggestions have been incorporated.
Author Response
We thank the reviewer for providing the valuable feedback to improve the manuscript.
Reviewer 3 Report
My comments have been basically addressed. However, details are not provided for the metabolic diseases accompanied with obesity in introduction.
Author Response
We are thankful to the reviewer for the concern. We have added the following text to the introduction to address and improve the manuscript:
"Apart from metabolic syndrome, overweight and obesity predisposes people to a range of adverse health consequences including endocrine disorders (e.g., advanced pubertal development, polycystic ovarian disease), cardiovascular disease (e.g., hypertension), respiratory symptoms including breathlessness and obstructive sleep apnea, and some malignancies [32]. Many recent studies [33-35] have indicate that metabolic syndrome is associated with increased risk for both atherosclerotic cardiovascular disease (ASCVD) and type 2 diabetes. Compared to normal persons, people with metabolic syndrome have at least a 2-fold increase in risk for ASCVD and about 5-fold Risk for type 2 diabetes in both men and women [36] In addition, diabetes is accompanied by microvascular disease, which is a common cause of chronic renal failure. The relationship between the metabolic risk factors and development of ASCVD is complex and certainly not well understood."